# A novel artificial intelligence-based approach for identification of deoxynucleotide aptamers

**Frances L. Heredia**[1], **Abiel Roche-Lima**[2], **Elsie I. Parés-Matos**[1]*

**1** Department of Chemistry, University of Puerto Rico-Mayagüez Campus, Mayagüez, Puerto Rico, United States of America, **2** Center for Collaborative Research in Health Disparities, University of Puerto Rico-Medical Sciences Campus, San Juan, Puerto Rico, United States of America

* elsie.pares@upr.edu

**Data Availability Statement:** The data is available as Heredia F. DNA/Aptamer dataset, Mendeley. 2020;1. doi: 10.17632/76jgjbgndr.1, and in GitHub at https://github.com/eipm-uprm/Aptamer-ML.

## Abstract

The selection of a DNA aptamer through the Systematic Evolution of Ligands by EXponential enrichment (SELEX) method involves multiple binding steps, in which a target and a library of randomized DNA sequences are mixed for selection of a single, nucleotide-specific molecule. Usually, 10 to 20 steps are required for SELEX to be completed. Throughout this process it is necessary to discriminate between true DNA aptamers and unspecified DNA-binding sequences. Thus, a novel machine learning-based approach was developed to support and simplify the early steps of the SELEX process, to help discriminate binding between DNA aptamers from those unspecified targets of DNA-binding sequences. An Artificial Intelligence (AI) approach to identify aptamers were implemented based on Natural Language Processing (NLP) and Machine Learning (ML). NLP method (CountVectorizer) was used to extract information from the nucleotide sequences. Four ML algorithms (Logistic Regression, Decision Tree, Gaussian Naïve Bayes, Support Vector Machines) were trained using data from the NLP method along with sequence information. The best performing model was Support Vector Machines because it had the best ability to discriminate between positive and negative classes. In our model, an Accuracy (A) of 0.995, the fraction of samples that the model correctly classified, and an Area Under the Receiving Operating Curve (AUROC) of 0.998, the degree by which a model is capable of distinguishing between classes, were observed. The developed AI approach is useful to identify potential DNA aptamers to reduce the amount of rounds in a SELEX selection. This new approach could be applied in the design of DNA libraries and result in a more efficient and faster process for DNA aptamers to be chosen during SELEX.

## Author summary

In this manuscript authors explain the development and validation of a novel artificial intelligence approach to support and simplify the early steps of the process from SELEX, to help discriminate binding between deoxynucleotide aptamers from those unspecified targets of DNA-binding sequences. The approach was implemented based on Natural

**Funding:** Research reported in this publication was supported by RCMI grant U54 MD007600 (National Institute on Minority Health and Health Disparities) from the National Institutes of Health (https://www.nimhd.nih.gov/programs/extramural/research-centers/rcmi/rcmi-grants.html). The content is solely the responsibility of the authors and does not necessarily represent the official views of the National Institutes of Health. The funders had no role in study design, data collection and analysis, decision to publish, or preparation of the manuscript.

**Competing interests:** The authors have declared that no competing interests exist.

**Abbreviations:** AI, Artificial Intelligence; DT, Decision Tree; GNB, Gaussian Naïve Bayes; LR, Logistic Regression; ML, Machine Learning; NLP, Natural Language Processing; SVM, Support Vector Machines; SELEX, Systematic Evolution of Ligands by Exponential enrichment; t-SNE, t-distributed Stochastic Neighbor Embedding.

Language Processing and Machine Learning. CountVectorizer, a Natural Language Processing method, was used to extract information from nucleotide sequences. Four Machine Learning algorithms (Logistic Regression, Decision Tree, Gaussian Naïve Bayes, and Support Vector Machines) were trained using data from the Natural Language Processing method along with sequence information. From these four trained machine learning algorithms, the best performance and selected model was Support Vectors Machines, because it had the best discriminatory metrics (i.e., Accuracy (A) = 0.995; AUROC (AU) = 0.998). In general, all models showed good metric results for predicting DNA aptamer sequences. The Machine Learning model complexity and difficult interpretation may hinder its application into the standard practice. For this reason, the development of a web-app is already taking place to facilitate the interpretation and application of the obtained results.

## Introduction

Aptamers are non-genomic, but biologically active single-stranded nucleic acid molecules, typically ranging between 10 and 100 nucleotides [1]. These short sequences can be designed to bind, with high affinity and specificity, to a broad spectrum of molecular targets, ranging from ions, small organic molecules to macromolecules such as proteins, viruses, and entire cells [2–8]. Aptamers assume a variety of shapes due to their tendency to form helices and single-stranded loops [9–11]. They are extraordinarily versatile and bind targets with high selectivity and specificity. Applications of aptamers in the field of medicine include diagnostic devices, therapeutic drugs, and antibody replacement and drug delivery systems [12–21]. Aptamers are of high interest to the pharmaceutical industry due to substantially lower production costs, shelf lives of years, and, in many cases, high target specificity [22]. Moreover, since aptamers are chemically synthesized, they can provide a reliable source of raw materials than antibodies that are secreted by cells.

The selection of a DNA aptamer by Systematic Evolution of Ligands by EXponential enrichment (SELEX) consists of a binding step by mixing a target and a library containing vast patterns of randomized DNA sequences, each with a common fixed-sequence primer region [23]. A separation step is needed to isolate multiple target–DNA complexes from unbound DNA, followed by separation of the complexes employing filtration or chromatography techniques, and an amplification step by PCR [24]. The DNA sequences obtained are re-used as new DNA aptamer enriched pools, followed by another series of selection steps, called a "round". After repeated rounds, the DNA aptamers in the pools are sufficiently enriched and ready to be sequenced and evaluated as aptamers by way of a binding assay [25]. SELEX may require 10–20 rounds, leading to an overall procedure that is complex and time-consuming [26].

The SELEX technique has several limitations, for example, it requires 10 to 20 rounds to be completed, but an increase in by-products can be found after seven or more rounds, and for protein and small molecule targets a decrease in affinity was found to occur after 5–6 rounds [27–29]. Multiple rounds of SELEX significantly bias the types of sequences [30]. Enrichment of unspecified binding of oligonucleotides during this aptamer selection process is often observed [31]. Most of the aptamers that have been published were manually selected, making the whole process of getting high affinity and specific aptamers time consuming [32]. Thus, a process requiring fewer rounds for aptamer selection is desirable.

Interest in the use of statistical methods in aptamer prediction approaches has grown lately. Computational techniques are simple, time-saving, cost-effective, and do not require specialized resources [33]. Aptamer's computational prediction methods have been carried out in two major categories: prediction based on interaction and prediction based on structure. Computational prediction models based on interaction, take into account the physicochemical, energetic and conformational properties of the aptamers. These models, while may not be very accurate, may shed some light on in-depth understanding of the mechanisms of interactions between aptamers and their targets, but cannot be applied to the SELEX pipeline to reduce the number of steps [34]. Computational prediction models based on structure folding tend to be more accurate, but their use is hampered by the dependency on the availability of homologous sequences [35].

For all these reasons, a novel Artificial Intelligence approach that includes Natural Language Processing and Machine Learning (ML) was developed to support aptamer–target interaction research with advanced computational tools. Our approach provides results that allow researchers to discriminate between the aptamer and non-aptamer sequences for efficient SELEX data analysis. This approach can be used at the early rounds of SELEX, to help distinguish between specific and unspecific binding sequences. Moreover, the consensus aptamer sequence can be selected at the final round of SELEX. When more data are available, this approach could be applied to obtain more reliable predictive models that can eventually reduce the number of rounds of SELEX for consensus aptamer sequence identification.

## Materials and methods

The workflow used in this paper (Fig 1) described our Artificial Intelligence (AI) approach. It included the Natural Language Processing (NLP) method for k-mer vectorization, along with feature selection, oversampling, machine learning training algorithms, and validation.

### Web scraping

The aptamers selected for this study were DNA aptamers with no modifications in neither the bases nor the backbone. The data for the DNA Aptamers was web-scraped from the online database Aptagen [36] using a python script. After downloading the data into a data frame, cleaning the data and removing sequences with modified bases and duplicates, the Dataframe included 238 unique aptamer sequences. The data for the DNA sequences was web-scraped from the Nucleic Acid Database (NDB) [37] using a tailored python script, while NDB reports both the 5' and the 3', only the 5' sequences were scrapped. After downloading the data, a tailored scripting was implemented for data wrangling (i.e., cleaning and removing sequences with modified bases and duplicates). The final Dataframe included a total of 4,885 of unique sequences. The codes for the python scripts in GitHub [38] and the raw data are available at Mendeley Data [39].

### Feature engineering

#### NLP Method–CountVectorizer

NLP techniques were used to transform the nucleotide sequences into k-mers numerical vectors to be used as input in the machine learning training algorithms. A CountVectorizer function, included in the SckitLearn library [40] was applied using n-grams (n = 6) as a parameter. The CountVectorizer algorithm used in the analysis is an enumeration algorithm, which counts the total occurrence of all possible k-mers (or n-grams) of a given length 'k' ('n'). *K*-mer counting involves counting the number of substrings that have length *k* in a string *S*, or a

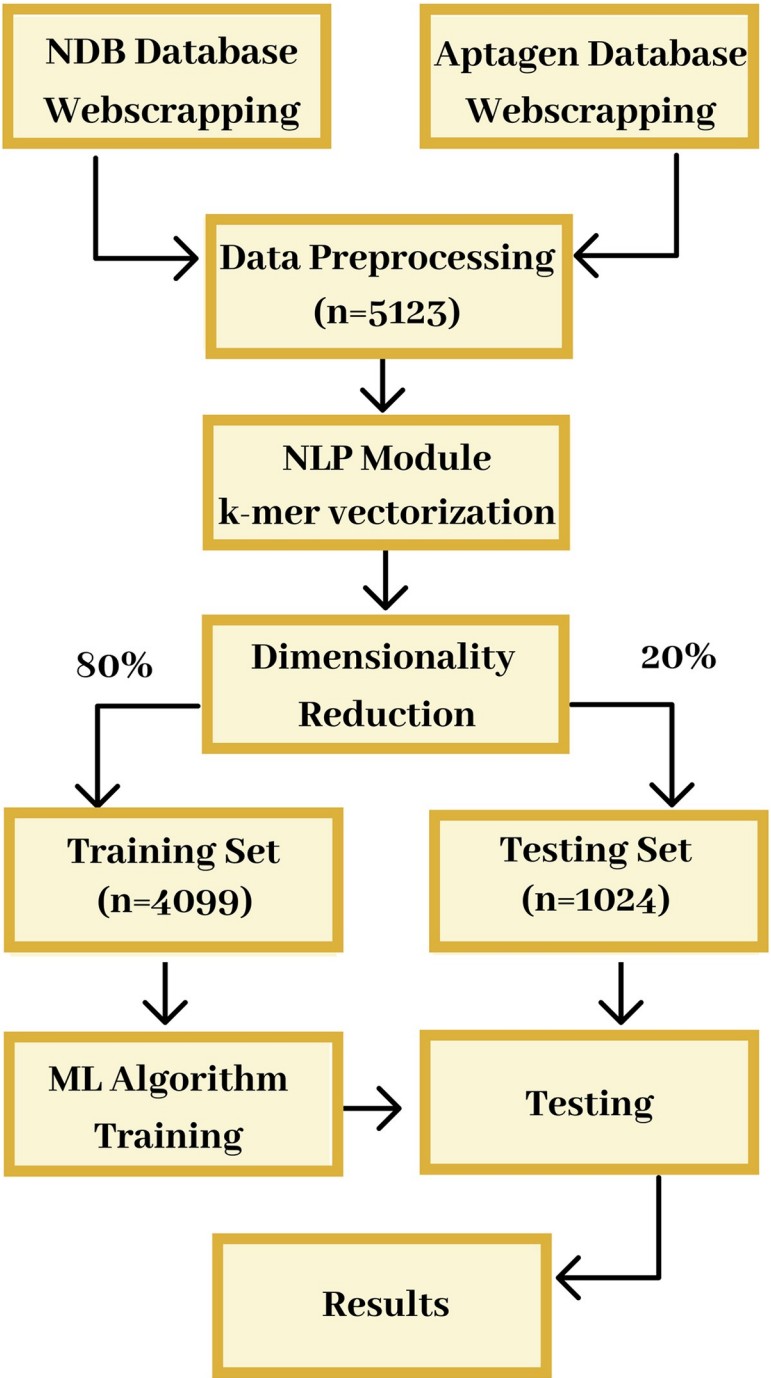

**Fig 1. Overview of the AI approach used to obtain a model for the classification of a sequence as an aptamer.** It included the extraction of nucleotide sequences from the Nucleic Acid Database (NDB) and Aptagen. The sequences were converted into 6-mer vectors using the NLP modules. Out of the 5,123 vectors created, only the top 2.5% were selected for modeling, in the reduction of dimensionality module. Then the data was split into a training set (80% of the data, n = 4,099) and test set (20% of the data, n = 1,024). Because of data imbalance in the training set, the underrepresented samples were weighted highly. ML algorithms were trained to develop the models using the selected features. The developed models were tested using cross-validation and validated using the test sets. **Fig 1** is also the Graphical Abstract.

set of strings, where $k$ is a positive integer. For any length k, there are $4^k$ combinatorically possible k-mers. The 'k' ('n') value was set to 6 because a previous study indicated that 6-mers performed better than k-mers of other lengths in target-aptamer identification [41]. The set of these 6-mers vectors contained numerical information that described the nucleotide sequences. They were used as features to train the ML algorithms and obtain the predictive models.

## Other computed features / variables

In addition to the 6-mer vectors, other features were also calculated for each nucleotide sequence. These features were sequence length, percentage of each base (Adenosine Percentage, Cytosine Percentage, Glutamine Percentage, and Thymine Percentage), AT ratio, CG ratio, purine ratio, and pyrimidine ratio.

## Dimensionality reduction

Only the top 2.5% most frequent features out of the 4,096 ($4^6$) originally generated were considered. It represented a total of 101 6-mers vectors. Before modeling, dimensionality reduction was performed on the remaining features using recursive feature elimination using logistic regression as the estimator [42]. The total number of features used to train the ML models was reduced to 33.

## ML classification–modeling

The data was split randomly with stratified sampling by sequence type to achieve a roughly equal proportion of DNA and aptamers, in 80% for training and 20% for testing sets. One of the biggest challenges found was the small number of aptamer sequences (as a sample), which meant an unbalanced target binary variable (i.e., 238 aptamers and 4885 DNA sequences). After testing undersampling and oversampling methods, such as SMOTE [43], the weights of the target variables were selected as a method to solve the unbalance data. The parameter CLASS_WEIGHT was set as '*balanced'* in each ML training algorithm to avoid undermining the models' predictability. Four of the most common supervised machine learning algorithms for classification were used to be trained with our training set (i.e., Logistic Regression, Decision Tree, Gaussian Naïve Bayes and Support Vector Machines). A set of metrics [44] was chosen for model performance comparisons including Accuracy, Specificity, Sensitivity and AUROC metric values, which are defined as follows:

- Accuracy is the fraction of samples that the model correctly classified and is defined as (TP+TN)/(TP+FP+FN+TN), where TP is True Positive, FP is False Positive, FN is False Negative, and TN is True Negative.

- Specificity is the ratio of samples that the model correctly classified as negative classes to all the negative samples, and is defined as *TN/(TN+FP)*.

- Sensitivity represents the ratio of samples that the model correctly classified as positives classes to all the positive samples, and is defined as *TP/(TP+FN)*.

- **Area Under the Receiver Operating Characteristics** (AUROC) is a probability curve where the true positive rate is plotted against the false positive rate, the area under this curve represents degree by which a model is capable of distinguishing between classes [45].

The confusion matrix is a tabular display of the samples by their actual and predicted class. Validations results using the testing set, as well as the confusion matrix for each model, were also computed and reported.

### Logistic Regression (LR)

For the Logistic Regression (LR) classifier [46], a Grid search was used to tune model hyperparameters, using a 5-fold cross-validation. The final classifier used an L2 penalty, a C value of 2, and an *lbfgs* solver.

### Decision Tree (DT)

Decision Tree (DT) classifiers have a comprehensible classification model that in many different cases, including balanced datasets, is highly accurate [47]. Each node in the tree specifies a test on an attribute, each branch descending from that node corresponds to one of the possible values for that attribute. Each leaf represents class labels associated with the instance. The final classifier used a squared root function to determine the maximum number of features.

### Gaussian Naïve Bayes (GNB)

Gaussian Naïve Bayes (GNB) classifiers are based on the Bayes Theorem [48]. This classifier assumes that the value of a particular feature is independent of the value of every feature. Naïve Bayes classifiers were chosen for this study because they need a small training sample to estimate the parameters needed for classification.

### Support Vectors Machine (SVM)

Support Vector Machines (SVM) is a supervised machine learning algorithm that can be used for classification of high dimensional data [49]. It uses a technique called kernel trick, where data points are placed above and below the classifying hyperplane. The data is transformed, and based on these transformations, it finds an optimal boundary between the possible outputs. Some benefits of the SVM is the capture of more complex relationships between the data points. Its disadvantage is that the training time is much longer and it is computationally intensive, and there is no probabilistic explanation for the classification. SVM can accurately deal with complex non-linear boundary models. A Grid search with 5-fold cross-validation was used to tune model hyperparameters: C, a hyper**parameter which** adds a penalty for each misclassified data point and gamma, a hyperparameter which controls the level of influence of a single training point has on the model. The final classifier used a C value of 10 and a gamma of 0.01.

### Validation and plots

All of the models were validated using a 5-fold cross-validation with accuracy as the scorer matrix. The cross-validation sets were generated from the initial dataset. The generated models and the confusion matrix were plotted for visual inspection. The symmetric correlation matrix was calculated and transformed into a heatmap to depict the relationship between all 6-mer sequences. Heatmaps were generated using the heatmap function in Seaborn [50]. Bar and Scatter plots were generated using the plot functions in Matplotlib [51].

### Characterization of the biological implications of the top 6-mers

DNA aptamer sequences and their structures were downloaded from Protein Data Bank (PDB) to understand the biological significance of the generated 6-mers. These sequences were used as input in a function that identifies the top 6-mers, according to Table 1. The structures where the 6-mers were found, were analyzed to understand their biological role.

**Table 1. Comparison of existing aptamer predictive studies.**

| Algorithm | Aptamer Dataset | No-aptamer Dataset | Some Features | Classifier | MCC |
|---|---|---|---|---|---|
| **This Study** | DNA aptamers (n = 238) | Protein binding DNA (n = 4885) | 6-mers for all sequences, Sequence features | Support Vector Machines | 0.896 |
| [35] | DNA/RNA aptamers (n = 159) Small Molecule Targets (n = 20) | Randomly paired aptamers to Small Molecule targets | 1,2-mers for aptamers, Physical-chemical properties of targets | Nearest Neighbors | 0.670 |
| [57] | DNA/RNA aptamers (n = 725) Protein Targets (n = 164) | Randomly paired aptamers to protein targets | 1,2-mers for aptamers, 1,2-mers for targets Physical-chemical properties of targets | Random Forest | 0.461 |

## Results and discussions

Explosive progress in high-throughput DNA sequencing has driven advances in analytical tools to identify base consensus motifs among subgroups of DNA sequences [52]. These sequence analysis tools can also be employed to identify patterns among non-genomic, yet functional, oligonucleotides called aptamers. Though aptamers are classified as non-genomic sequences, tools built for genomic sequence analysis can still be useful, as demonstrated in studies with RNA aptamers [53,54]. Our AI approach, that leverages NLP and ML techniques, is developed to classify and discriminate DNA aptamer sequences from genomic DNA sequences. For this study, the Aptagen and NDB databases are web-scrapped to retrieve all sequences of published DNA aptamers and protein binding DNA sequences, respectively.

### Baseline characteristics

From 5,123 sequences retrieved, 4,885 are protein-binding DNA sequences from NDB, and 238 are DNA aptamer sequences from Aptagen. A group of features/variables is initially determined by the NLP method (i.e., 6-mers vectorization function) where a total of 33 6-mer features are used after dimensionality reduction. Other features/variables chosen for the DNA and aptamer sequences used in this study are shown in Table 2. The DNA sequences have an even distribution of the bases, while in the aptamer sequences the distribution of the bases was skewed towards the thymine and guanine residues. The AT and CG ratios are calculated for both types of sequences. In the DNA sequences, the AT and GC ratios are 0.5 on average. Meanwhile, in the aptamer sequences, the CG and AT ratios are 0.54 and 0.45, respectively, suggesting that aptamers are lightly more stable than DNA sequences. As above mentioned, 4,099 (80%) and 1,024 (20%) sequence data are randomly assigned to the training and testing

**Table 2. Characteristics and other information of the other features/variables that corresponds to DNA and aptamer sequences (data are presented as Mean ± SD).**

| Variables | Overall n = 5123 | DNA Samples n = 4885 | Aptamer Samples n = 238 | P-value | Training Set n = 4099 | Testing Set n = 1024 | P-value |
|---|---|---|---|---|---|---|---|
| **Adenosine Percentage** | 24.1±10.4 | 24.3±10.4 | 20.8±8.4 | < 0.001 | 24.2±10.4 | 24.1±10.4 | 0.900 |
| **Cytosine Percentage** | 24.7±10.3 | 24.9±10.4 | 21.5±8.4 | < 0.001 | 24.7±10.3 | 24.9±10.4 | 0.613 |
| **Glutamine Percentage** | 26.6±11.3 | 26.4±11.2 | 32.4±12.3 | < 0.001 | 26.7±11.3 | 26.7±11.2 | 0.950 |
| **Thymine Percentage** | 24.4±10.5 | 24.4±10.6 | 24.7±8.5 | 0.600 | 24.4±10.5 | 24.3±10.5 | 0.727 |
| **AT Ratio** | 0.48±0.14 | 0.49±0.15 | 0.40±0.10 | 0.001 | 0.49±0.15 | 0.48±0.15 | 0.739 |
| **CG Ratio** | 0.51±0.15 | 0.51±0.15 | 0.50±0.10 | 0.007 | 0.51±0.15 | 0.52±0.14 | 0.691 |
| **Purine Percentage** | 0.51±0.11 | 0.51±0.11 | 0.50±0.10 | <0.001 | 0.51±0.11 | 0.52±0.15 | 0.956 |
| **Pyrimidine Percentage** | 0.49±0.10 | 0.49±0.11 | 0.50±0.10 | <0.001 | 0.49±0.11 | 0.51±0.11 | 0.885 |

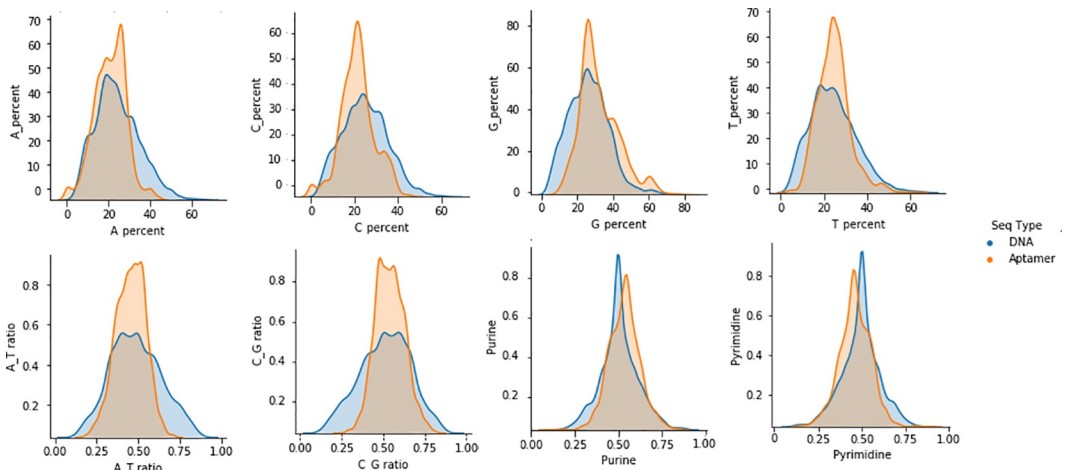

**Fig 2. Plot of the DNA vs. aptamer features.** Variables are against themselves to show the distribution. The blue area depicts DNA sequences while orange area depicts aptamers sequences.

sets, respectively. After calculating the p-values, it is determined that the characteristics are similar between the training and testing set for these variables/features.

A pair plot (Fig 2) represents a visual representation of the relationship between these other features/variables in the dataset. It is built based on the density plot and the scatter plot. In the density plot, the diagonal shows the distribution of a single feature. At the bottom right corner, density plots for purine and pyrimidine distribution show a leptokurtic shape, with a mean around 50. For other features, aptamers have shown a leptokurtic distribution, while DNA features have a Gaussian distribution. Scatter plots above and below the density plots, show the relationship (or lack thereof) between the features of the two nucleotide types. These additional plots suggest that a single feature/variable is not enough to discriminate between DNA aptamers and DNA sequences without the use of a more sophisticated model.

## Feature/variable exploration

The t-distributed Stochastic Neighbor Embedding (t-SNE) is plotted to further explore the generated data. This algorithm for dimensionality reduction is particularly well suited for the visualization of high-dimensional datasets [55]. The t-SNE algorithm calculates a similarity measure between pairs of variables in the high dimensional space and the low dimensional space. It then tries to optimize these two similarity measures using a cost function. In this way, t-SNE plots the multi-dimensional data to a lower dimensional space and attempts to find patterns in the data by identifying observed clusters based on the similarity of data points with multiple features. In Fig 3, DNA sequences are represented by the blue dots, while aptamer sequences are represented by the orange dots. The plots show that DNA sequences can be clustered into a group, while DNA aptamers are more scattered and more different from each other. Also, some aptamer samples fall into the DNA cluster. This result suggests that these samples are very similar to DNA samples and, therefore, it would be challenging for a model to predict. Although some DNA samples fall outside of their cluster, they are far apart from the aptamer samples and could be predicted by a model. However, it is important to note that the distances between points are relative because the algorithm is non-linear, the distances shown on the x- and y-axis have no direct interpretation.

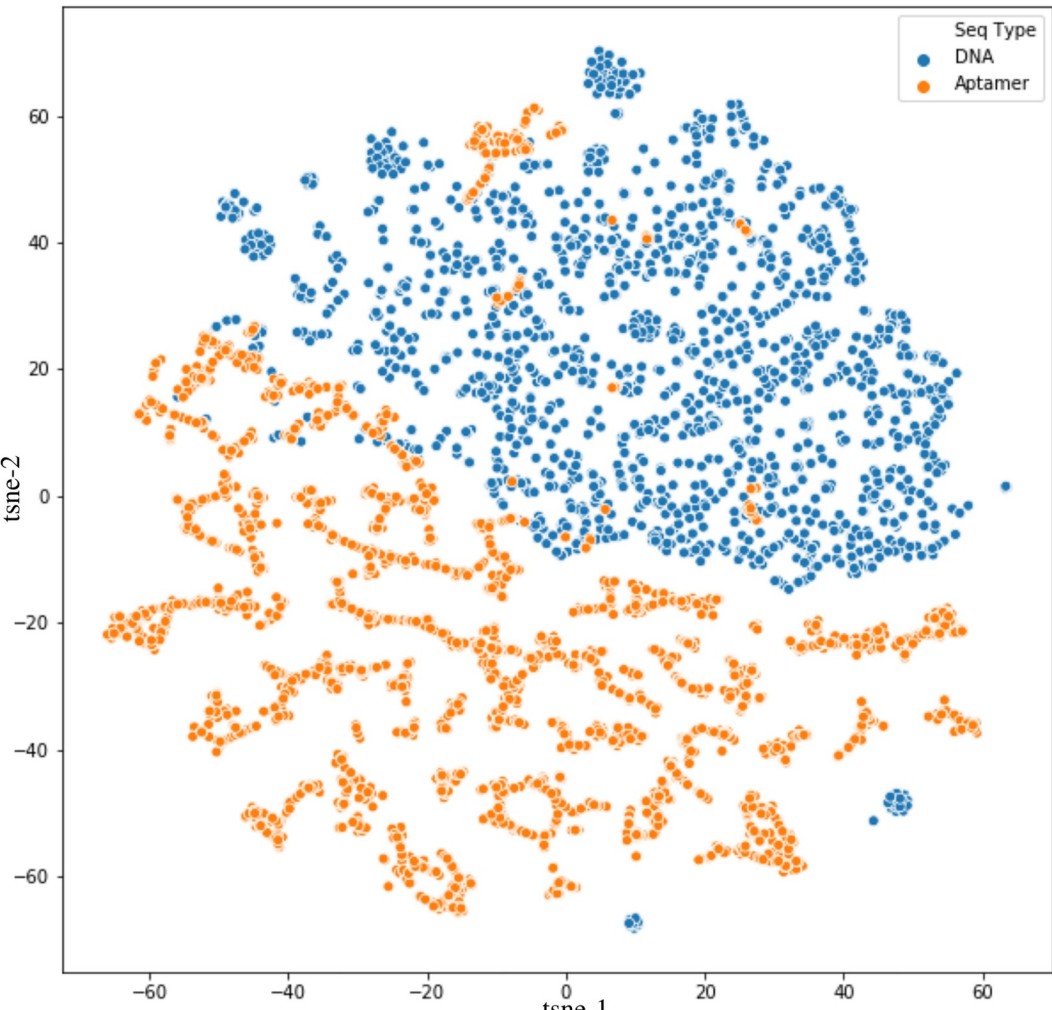

**Fig 3. t-SNE of the Dataframe colored by sequence type.** Blue dots depicts genomic DNA sequences, while orange dots depicts DNA aptamer sequences.

In Fig 4, the observed occurrences of the 6-mer are plotted. They are calculated and normalized for each of the oligonucleotides. It is evident from this figure that the distribution of the 6-mers varies greatly from DNA to aptamers and that the 6-mers with high GT content are more frequent in aptamers. In Fig 5, 6-mer content was compared in an all-versus-all pairwise fashion, to determine correlation coefficients (CC) of 5,100 comparisons in total (including comparisons between the same sequences). The CC values represent how likely two 6-mers to be present within one sequence are. The darker the color, the more occurrences of these two 6-mers being current within one sequence. In the heatmap, a darker, bluer color denotes a higher CC value, closer to 1. Lighter white, colors indicate CC values closer to 0, shows 6-mer pairs unrelated.

## ML algorithm performances and final models

The training data set was used to train the ML algorithms. The computed metric values (i.e., Accuracy, Sensitivity, Specificity, and AUCROC) can be seen in Table 3. For LR obtained model, the accuracy and AUROC are 96.3% and 0.988, respectively. Fig 6 shows information

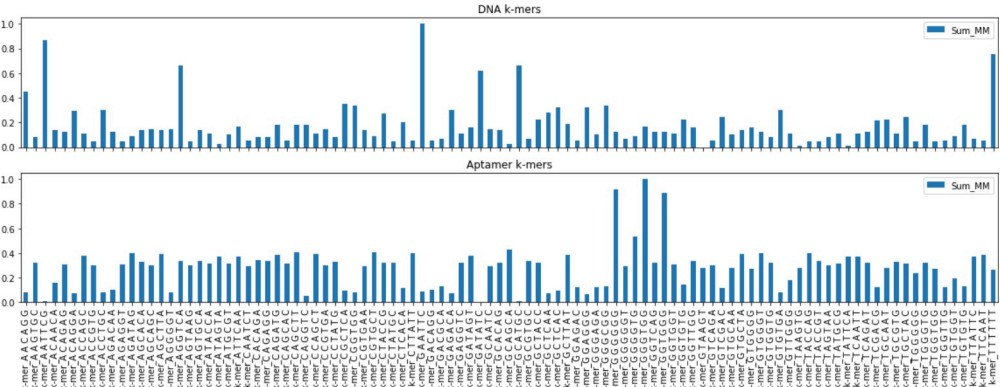

**Fig 4. Bars plot of selected 6-mers and their normalized distribution.** Top graph: distribution of chosen 6-mers in genomic DNA sequences. Bottom graph: distribution of chosen 6-mers in DNA aptamer sequences.

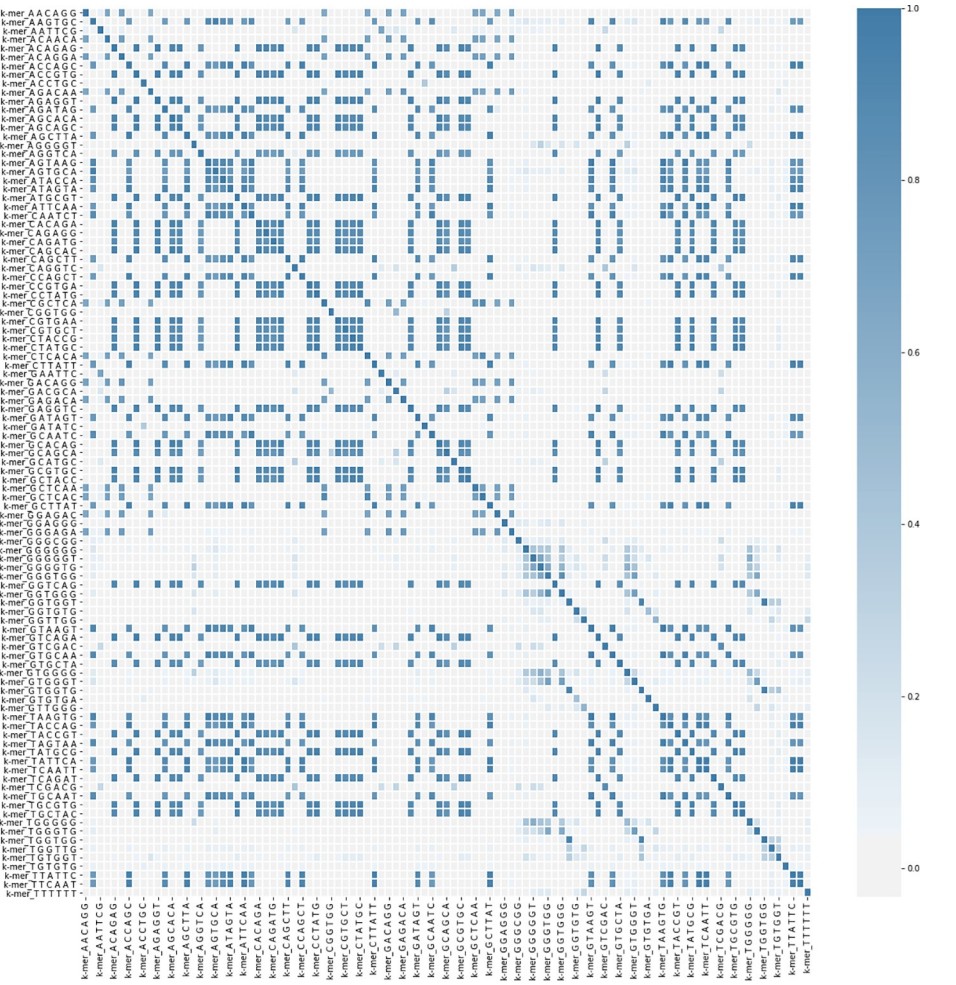

**Fig 5. The heatmap of features correlation.** The blue diagonal represents a correlation factor equals to one. Blue color means a positive correlation, while white color means no correlation.

**Table 3. Classifiers' predictive performance in the testing set.**

| Classifier | Accuracy | Sensitivity | Specificity | AUROC |
|:---:|:---:|:---:|:---:|:---:|
| LR | 0.963 | 0.999 | 0.893 | 0.988 |
| SVM | 0.992 | 0.997 | 0.878 | 0.998 |
| DT | 0.990 | 0.998 | 0.805 | 0.918 |
| GNB | 0.917 | 0.919 | 0.878 | 0.926 |

about the confusion matrix and plots for each ML algorithm. Fig 6A corresponds to LR. DT algorithm is also used. It is recommended when the dataset is small or when the data is imbalanced [56]. As can be seen in Table 3, when the DT is trained, the accuracy of the obtained model increased to 99.0%, but the AUROC decreased to 0.918, showing that the model is more suitable to predict DNA sequences instead of aptamer sequences (confusion matrix and plot can be seen in Fig 6B). GNB is another ML algorithm that is trained with our data set (Fig 6C shows the plots and confusion matrix). The accuracy of the model obtained with GNB is lower

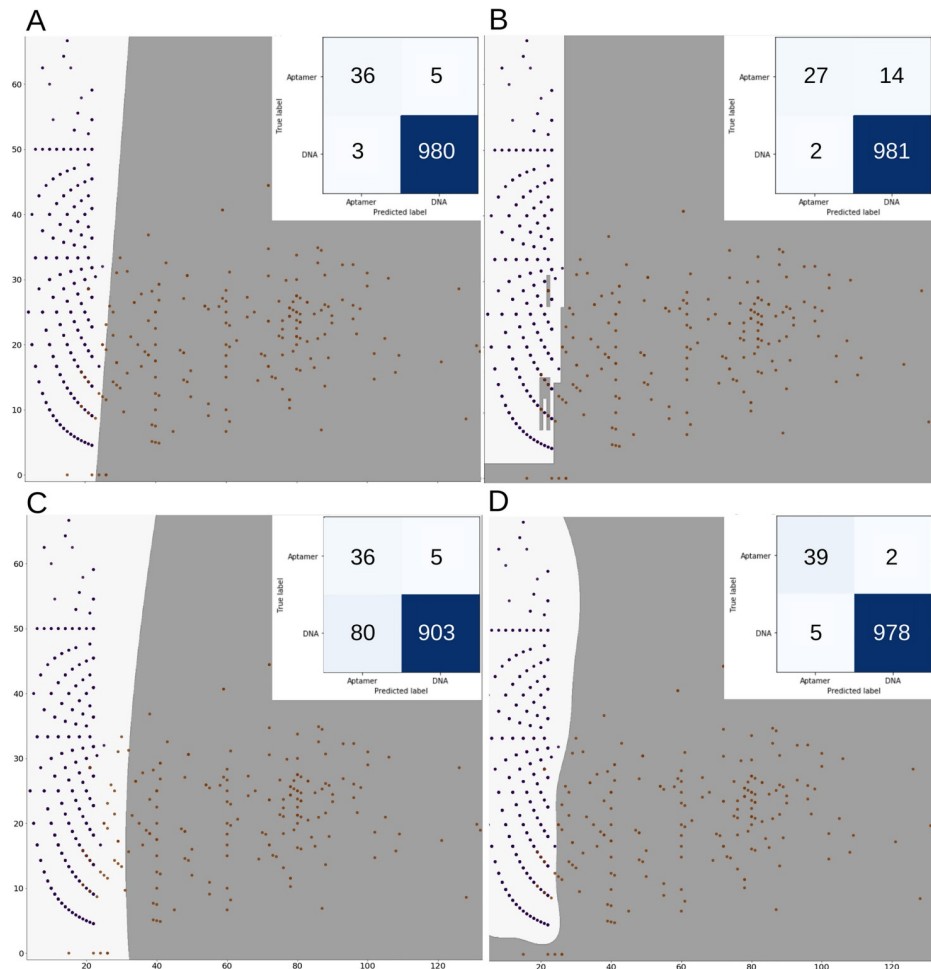

**Fig 6. Scatter plots of each ML model.** DNA aptamer sequences are shown as orange dots and DNA sequences are shown as dark blue dots. The insert shows the confusion matrix of each model. (A) Logistic Regression, (B) Decision Tree Classifier, (C) Gaussian Naïve Bayes and (D) Support Vector Machines. The light gray area is the boundary for predicted DNA sequences and the dark gray area is the boundary for predicted DNA aptamer sequences.

than the previous models as can be seen in Table 3. SVM is also trained to develop a model for aptamers. The results in Table 3 show that the final SVM model has the highest metric values for accuracy and AUROC (i.e., 99.2% and 0.998, respectively). It suggests that SVM model has the best discrimination between DNA and Aptamer sequences, as can be seen in Fig 6D. In general, all models show a high AUCROC metric value for predicting aptamer sequences, as can be shown in Fig 7.

## Comparison with other reported ML algorithms

Other ML algorithms have been used to obtain models to predict aptamer sequences. One of such algorithms uses sequences of DNA and RNA aptamers from the now defunct Aptamer Base Database along with the physical-chemical properties of the small molecule targets they bound to. Using a Nearest Neighbors algorithm, they managed to obtain a Matthews Correlation Coefficient (MCC) metric value of 0.670 and the main predictive features in their study were related to the electrostatic and chemical descriptors of the target molecules and not the aptamers themselves [35]. The second reported ML approach also employed sequences of DNA and RNA aptamers from the Aptamer Base Database along with the protein targets they bound [57]. For this algorithm, they calculated the 1-mers and 2-mers for both the aptamer and protein sequences, and physical-chemical properties of the proteins. Using a Random Forest algorithm, they obtained a model with MCC metric value of 0.461 to predict the top aptamer related to the target. These algorithms use aptamers sequences randomly paired to the targets as the negative data to train on. To the best of our knowledge, our reported models are the only ones capable of discriminating aptamer sequences from non-aptamer sequences. In addition, our study identifies which sequence features make good candidates for aptamers. To compare our approach with the results from the other studies (i.e., [50] and [51]) we computed the MCC metric for our best model, this is SVM. As can be seen in Table 1, we obtain the best MCC value.

## Characterization of the biological implications of the top 6-mers

The feature ranking method is used to identify the optimal features required for high accuracy in the ML algorithms. Out of the 110 features that are used as input in the principal component

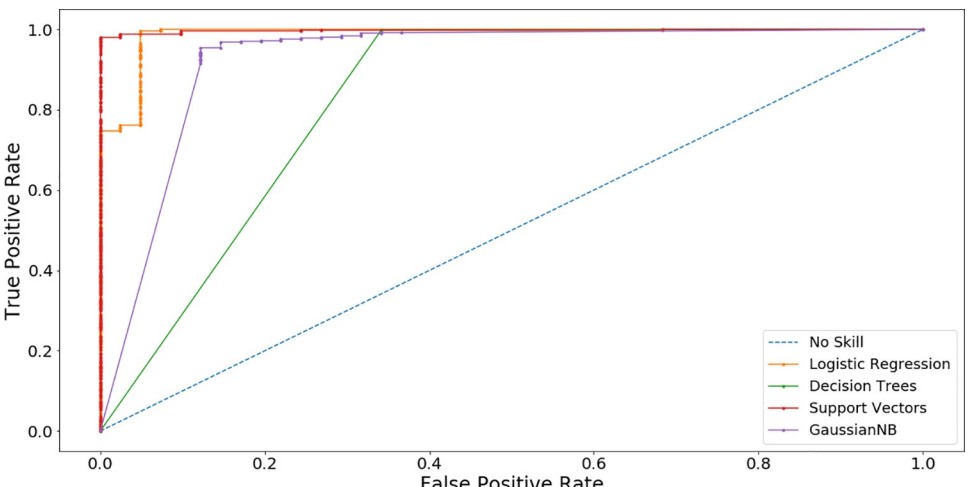

**Fig 7. Receiver-operating characteristic curve by machine learning model.** The closer the ROC curve is to the upper left corner, the higher the overall accuracy of the test.

**Table 4. Top predictive features (6-mers), their reported structural function and contribution.**

| Predictive Sequence | Reported Structural Function | References |
|---|---|---|
| TGG TGG | A section of a G-quadruplex that interacts with Thrombin | [58] |
| TGG GGG | A section of a hairpin loop. | [10,59] |
| GGG GTG | A section of a hairpin loop that interacts with Thrombin and VEGF. | [60] |
| GGT TGG | Complimentary chain in the G-quadruplex the two thymine (T) residues interact with Thrombin. | [58,60,61] |
| GGG GGG | A small section of the hairpin loop that interacts with HIV-1 reverse transcriptase. | [62] |

analysis, the top 33 are 6-mers vectors. According to Table 4, the 6-mers with the highest relevance across all ML models were **TGG TGG**, **TGG GGG**, **GGG GTG**, **GGT TGG**, **GCA CAG** and **GGG GGG**. From the top 6-mers identified, five are found as structures in PDB (Fig 8) and have been involved in protein binding. Three of these five 6-mers are found in a hairpin motif, and two are found in a G-quadruplex arrangement.

## Aptamer predictive sequences

After the SVM model was generated, the more related sequences from these 6-mers (features) used to predict the DNA aptamer sequences are extracted. In the DNA sequences of all living organisms, 6-mers are short recurring elements. Within genomic DNA, due to their functional importance, these elements are both conserved and diverged across species, making these 6-mer patterns suitable for species identification. 6-mers may be part of the core segment of transcription factor binding sites or regulatory elements that participate in protein binding and gene regulation in different subregions of the genome [63]. Given the significance of 6-mers in genomic DNA, it could be assumed that 6-mers found in aptamers would also hold some biological or structural importance. To determine the biological importance of those 6-mers of six base pairs (bp) length, the top 5 predictive 6-mers are compared to the elucidated aptamer structures from the PDB database (Table 4). Some applications found on these 6-mers are described in Table 5. It is important to highlight that while 6-mers may have functional or structural importance in these aptamers, it is too premature to conclude that 6-mers always have biological significance. There are very few DNA aptamers that have been studied, from a structural perspective. Especially there are only 16 unique structures of DNA aptamers deposited in the PDB.

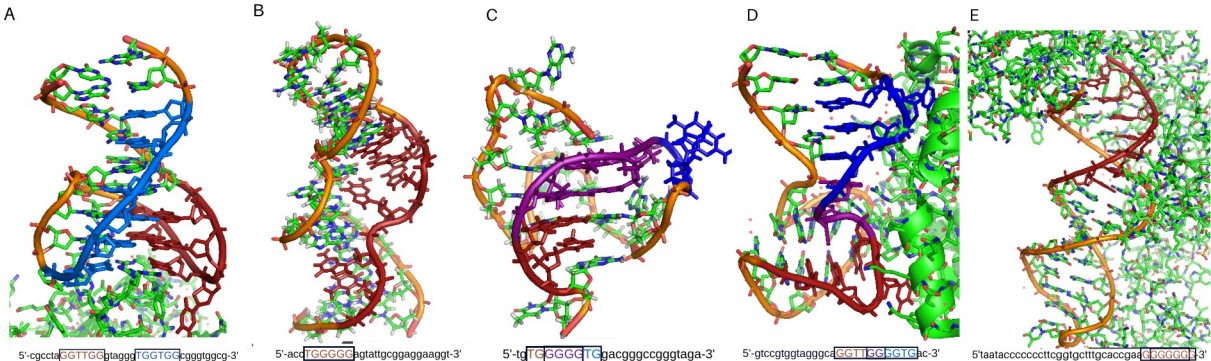

5'-cgcctaGGTTGGgtagggTGGTGGcgggtggcg-3'     5'-accTGGGGGagtattgcggaggaaggt-3'     5'-tgTGGGGGTGgacgggccgggtaga-3'     5'-gtccgtggtagggcaGGTTGGGGTGac-3'     5'taataccccccttcggtgctttgcaccgaaGGGGGG-3'

**Fig 8. Structures of elucidated Aptamers with their sequences.** The identified 6-mers are highlighted in red and blue, in the sequence and structure, the area where the 6-mers overlap is colored in purple. (**A**) NU172 Aptamer PDB:6GN7; (**B**) AMP Aptamer PDB: 1AW4; (**C**) V7t1 VEGF Aptamer PDB: 2M53; (**D**) HD22 Aptamer PDB: 4I7Y and (**E**) HIV-1 RT Aptamer PDB:5D36.

**Table 5. Identification of the top 6-mers on elucidated aptamer structures.**

| Aptamer name | Description |
|---|---|
| NU172 | The crystal structure of NU172 is shown in Fig 8A. This DNA aptamer was designed to bind Thrombin and has a high potency as an anticoagulant [58]. NU172 contains two of the top predictive 6-mers **GGT TGG** and **TGG TGG**. This structure has a chair-like anti-parallel fold, where the 6-mer **GGT TGG** forms G-tetrad type I and II. The second 6-mer **TGG TGG** is part of a TGT loop that surrounds the G-tetrad. This TGT loop is highly flexible and different from other TGT loops found in another DNA aptamer sequences. |
| AMP | The NMR structure of AMP is shown in Fig 8B. This DNA aptamer binds AMP as well as adenosine with an affinity shown to be 6 μM [10]. It has the 6-mer sequence **TGG GGG**, as one of two highly conserved guanine-rich regions. The **TGG GGG** sequence is part of two AMP-binding sites, which are located in the minor groove of the DNA helix. The **TGG GGG** also adapts a complex with the major groove centered about the adjacently bound AMP molecules. |
| V7t1 | The NMR structure of V7t1 is shown in Fig 8C. This DNA aptamer binds to $VEGF_{165}$ and $VEGF_{121}$, the two most abundant VEGF isoforms, with $K_D$ values at a very low nanomolar concentration [64]. V7t1 comprises several G-rich regions and folds into a G-quadruplex. It contains the top predictive 6-mers **GGG GTG** and **TGG GGG** that overlap in the sequence GTGGGGGTG. These nucleotides are numbered as G2–G10. Loop regions comprise a non-residue propeller-type loop between G6 and G7, a T9-G10 D-shaped loop connecting outermost residues G8 and G11 within the same strand. The DNA backbone is in an extended conformation in the G6-G8 tract, which causes displacement of G6 and G8 from what is considered their ideal stacking position. The DNA strand is connecting residues G8 and G11 of the outer G-quartets within the same column of a G-quadruplex core. G11 adopts anti-conformation along with its glycosidic bond, and G7-G8 and G11 segments can be considered parts of two DNA strands oriented in a parallel fashion. |
| HD22 | The crystal structure of HD22 is shown in Fig 8D. This DNA aptamer also binds to Thrombin and exhibits a substantially high negative charge density compared to other thrombin's aptamers, thus strengthening its specificity for target recognition [61]. It is also a bimodular aptamer with respect to a double helix and a G-quadruplex. This aptamer has the top predictive 6-mers **GGG GTG** and **GGT TGG** that overlap in a single sequence as GGTTGGGGTG. Its bases are numbered as G17–G25. This region of the aptamer is part of the duplex structure, and it is organized into a G-tetrad capped by the Thy18-Thy19 on one side. Interaction between HD22-27 and Thrombin involves numerous residues, including molecules such as Thy18, Thy19, Gua20, Gua23, Thy24, of the aptamer and segments 89–101, 230–245 of Thrombin. Hydrophobic contacts, mainly involving loop residues Thy18 and Thy19, also contribute to the stability of the complex. A further anchorage is produced by Thy24, which bulges from the duplex region of the nucleotide into a protein pocket where it is mainly involved in polar contacts. |
| HIV-1 RT | The crystal structure of HIV-1 RT is shown in Fig 8E. This DNA aptamer binds to HIV-1 reverse transcriptase with ultra-high affinity [62]. It has two repeats of the 6-mer **GGG GGG**, numbered as G27–G32, as part of the primer duplex strand. The conformational analysis of this aptamer suggest that base pairs conformation conforms into a B-form geometry. Nucleotides 28–33 can interact with crucial amino acid residues located in the p66 finger as well as in the palm and thumb subdomains of HIV-1 reverse transcriptase. |

## Conclusions

Aptamer screening efforts via SELEX could be defined as an obscure series of *in vitro* experiments, lacking the principles of a binding motif design in a screening library. In the absence of these principles, large random sequences have to be scanned first to recognize potential aptamers for a given target. In this work, an AI-based approach, based on NLP and ML, was developed to predict if a given sequence is an aptamer. It uses an NLP method to convert DNA sequences into numerical smaller representations (i.e., features) and ML to obtain a predictive model to classify a sequence as an aptamer or a genomic DNA sequence. The use of the best model examined in this paper to predict aptamers is promising to improve SELEX protocols and accelerate the rate of aptamer development. This new approach may allow DNA sequences resulting from the first-round of SELEX to be pre-selected as potential aptamers for the second-round of SELEX by eliminating non-specific binding sequences. This analytical step

could reduce the number of SELEX rounds required to produce a good aptamer. Based on these comparative studies, new screening libraries could be developed to overexpress the promising 6-mers found or intentionally excluding those features that are not indicative of a DNA aptamer. This approach will allow us to identify aptamers faster and more precisely, so more aptamers can be generated in the future.

One of the limitations of this work is that the examined methods considered all aptamers to be the same, and does not take into consideration the different binding strengths of the aptamers and the different types of binding targets (i.e., proteins, small molecules, whole cells). As more aptamer data becomes available, new studies could be done taking into account those differences. Also, this model has not been validated through experiments. Thus, future studies could include testing the model in a SELEX experiment to better address its viability. Future works include improving the quality of the models using more data, when this type of data becomes available. Finally, ML model complexity and difficult interpretation may hinder its application into the standard practice. For this reason, the development of a web-app is already taking place to facilitate the interpretation and application of the obtained results.

## Acknowledgments

Authors want to thank Luis E. Vázquez-Quiñones, professor of the School of Sciences and Technology of the Universidad Metropolitana-Ana G. Méndez, for his comments and suggestions during the writing of this manuscript.

## Author Contributions

**Conceptualization:** Frances L. Heredia, Elsie I. Parés-Matos.

**Data curation:** Frances L. Heredia, Abiel Roche-Lima.

**Formal analysis:** Frances L. Heredia.

**Funding acquisition:** Abiel Roche-Lima.

**Investigation:** Frances L. Heredia, Abiel Roche-Lima, Elsie I. Parés-Matos.

**Methodology:** Frances L. Heredia, Elsie I. Parés-Matos.

**Project administration:** Elsie I. Parés-Matos.

**Resources:** Frances L. Heredia.

**Software:** Frances L. Heredia.

**Supervision:** Abiel Roche-Lima, Elsie I. Parés-Matos.

**Validation:** Abiel Roche-Lima, Elsie I. Parés-Matos.

**Visualization:** Frances L. Heredia.

**Writing – original draft:** Frances L. Heredia, Elsie I. Parés-Matos.

**Writing – review & editing:** Abiel Roche-Lima, Elsie I. Parés-Matos.

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
