## [Decision Letter · Decision Letter 0]

27 Nov 2020

Dear Mrs. Pares-Matos,

Thank you very much for submitting your manuscript "A novel artificial intelligence-based approach for identification of deoxynucleotide aptamers" for consideration at PLOS Computational Biology.

Your manuscript has been reviewed by a team of expert scientists, who while appreciating the general idea, have detected many flaws concerning the quality of data and results, and also the accessibility of your manuscript, too technical for the standards of PLOS Computational Biology.

In conclusion,  we strongly suggest you to move the manuscript to a more technical journal, alternatively,

you are required to respond point by point to all the referee questions/comments, and to strongly revise the paper, submitting a new version, more accessible for a broader audience.

We cannot make any decision about publication until we have seen the revised manuscript and your response to the reviewers' comments. Your revised manuscript is also likely to be sent to reviewers for further evaluation.

Sincerely,

Eleonora Alfinito, Ph.D.

Guest Editor

PLOS Computational Biology

Florian Markowetz

Deputy Editor

PLOS Computational Biology

Dear Authors,

your manuscript has been reviewed by a team of expert scientists, who while appreciating the general idea, have detected many flaws concerning the quality of data and results, and also the accessibility of your manuscript, too technical for the standards of PLOS Computational Biology.

In conclusion, I strongly suggest you to move the manuscript to a more technical journal, otherwise,

you are required to respond point by point to all the referee questions/comments, and to strongly revise the paper, submitting a new version, more accessible for a broader audience.

Reviewer's Responses to Questions

**Comments to the Authors:**

Reviewer #1: - abstract: you mention accuracy and auroc parameters, this is too technical and at this point these quantities are undefined and may be obscure to the general audience. Find a better way to present that information.

- line 111: provide a reference for the previous study

- line 139: provide a definition of Accuracy, Specificity, Selectivity and AUROC metric

- line 170: provide a definition of C, gamma

- line 175: provide a definition for the 'confusion matrix'

- line 286: table 1, too many digits in mean values taking into account the error.

- line 329: fig. 2: I cannot read the vertical axis of the plots even zooming in. Better reduce the number of shown cases to most representative ones.

- line 351: fig. 6: what is meaning of grey shaded area ?

Reviewer #2: The authors present a novel artificial intelligence approach for identifying deoxynucleotide aptamer from DNA sequences using a combination of natural language processing and machine learning. To train the model, the DNA sequence data were retrieved from the public database, NDB and Aptagen. NLP method (i.e.,CountVectorizer) was used to extract features from the nucleotide sequence. Four machine learning models, i.e., Logistic Regression, Decision Tree, Gaussian NaïveBayes, Support Vector Machines were trained on data from which the authors identified Support Vector Machine as the best performing model. They further correlated some of the predictive features to 3D structures and demonstrated their important functional roles in biology. Overall, I found the paper interesting, easy to read and can have potential impacts on facilitating aptamer design. The paper is good, but several points need to be addressed to further improve its impact and clarify to the audience as well as its overall presentation.

1. I think the introduction should be improved. While the section provides excellent background on aptamer biology and SELEX, it does not mention any previous works using computational approaches for aptamer prediction, including existing 2D/3D structural approaches. A brief survey of the field on the computational side may be necessary.

2. In the training data, the DNA sequences (negative samples) were substantially shorter than the aptamer sequence. In the pairwise scatter plot in figure 2, even just the length along is sufficient to separate aptamer from non-aptamer sequence. Clear decision boundaries can also be drawn for other features. I’m wondering if this alone gives good model performance. I think it would be more convincing if the authors could train/test on non-aptamer DNA sequences with similar sequence/length to the aptamer sequence or perhaps using randomized sequences to make sure the model is non-trivial and could be applied for harder/real-world problem.

3. Along the same line, I think it would be more impactful if the authors could also develop a machine learning model to predict positive aptamer sequences given a binding target, not simply just reducing the candidate pool.

4. In figure 4, the profile of DNA vs aptamer sequence shows a very different distribution. However, these features were pre-selected based on the frequency not by the machine learning models. Therefore, it is unclear to me how these features contribute to the model performance and their relevance to the developed models. I would recommend perhaps training a regularized linear model such as elastic net to see if some of these k-mers features could be picked up by the model.

5. How were the top 33 k-mer ranked/identified? The principal component analysis does not predict feature importance. Also, how were the top predictive features in table 4 determined? It is a bit confusing because the input to the model seems to be PCA components according to Figure 1.

6. How was the PCA analysis look like for the data? Although PCA was used for feature reduction, tSNE was shown instead.

7. In table 3, the authors compared their machine learning model performance to several others existing machine learning approaches on aptamer predictions. However, difference training/testing dataset were used. While informative, I think it will be more relevant if the same dataset could be used for comparison.

8. What is the scatterplot in figure 6? The x axis shows aptamer, y axis shows DNA but each dot could be either aptamer or DNA.

9. The conclusion is quite brief. The authors could maybe put their approach into broader content to see how their approach could be developed for aptamer design. Perhaps also compare/contrast to existing approaches, such as those outlined in Table 3 to showcase the unique value/limitation of the approach.

10. There is no implementation/code to the model. Could the authors provide perhaps with the github page along with training/testing dataset?

11. Figure quality should be substantially improved and revised for better presentation. The font in figure 2 is quite small and is barely readable.

12. Line 202, “Error! Reference source not found”.

13. Line 253, AUROC are 96.3% and 0.98.

Reviewer #3: The paper proposes an approach to simplify SELEX early steps based on Natural Language Processing (NLP) and Machine Learning (ML). The method helps identify possible aptamers from those non-aptamer sequences. The performance of four ML algorithms, including Logistic Regression, Decision Tree, Gaussian Naïve Bayes, and Support Vector Machines, were analyzed and compared. The idea of an AI-based approach for identifying aptamers is interesting, and five 6-mers having high relevance with aptamers were identified by ML algorithms.

Comments and local corrections:

(1) Data sources. The work used 4,885 protein-binding DNA sequences and 238 aptamer sequences as datasets. These two types of data differ significantly in sequence length. Is it suitable to use them as a dataset, and what the reason for this choice?

(2) In Table 1, there are two p-values; what is the difference between them? Why do they have such a big gap?

(3) In the given URL, we could not find the codes for the python scripts.

(4) Page 5, in the caption of Figure 1, we think the following data should be examined: the number of vectors 4080? and the number of the training set is 7,816? And the test set is 1,954?

(5) Page 6, line 111-112, the references of the previous study in the sentence “…as set to 6 because a previous study indicated that 6-mers performed better than k-mers of other …” did not cite.

(6) Page 10, line 202, Error! Reference source not found.

(7) The expression of some terms is inconsistent, and this may confuse the readers. For instance, DT or DTC for Decision Tree.

**Have all data underlying the figures and results presented in the manuscript been provided?**

Reviewer #1: Yes

Reviewer #2: Yes

Reviewer #3: Yes

PLOS authors have the option to publish the peer review history of their article (what does this mean?). If published, this will include your full peer review and any attached files.

Reviewer #1: No

Reviewer #2: No

Reviewer #3: No
---

## [Decision Letter · Decision Letter 1]

2 Mar 2021

Dear Frances L. Heredia, Abiel Roche-Lima, Elsie I. Pares-Matos,

Thank you very much for submitting your manuscript "A novel artificial intelligence-based approach for identification of deoxynucleotide aptamers" for consideration at PLOS Computational Biology. As with all papers reviewed by the journal, your manuscript was reviewed by members of the editorial board and by several independent reviewers. The reviewers appreciated the attention to an important topic. Based on the reviews, we are likely to accept this manuscript for publication, providing that you modify the manuscript according to the review recommendations.

Sincerely,

Eleonora Alfinito, Ph.D.

Guest Editor

PLOS Computational Biology

Florian Markowetz

Deputy Editor

PLOS Computational Biology

[LINK]

Reviewer's Responses to Questions

**Comments to the Authors:**

Reviewer #1: The authors have answered my questions in a satisfactory way. The revised draft may be published.

Reviewer #2: Overall, the revised manuscript has been improved from the previous version.

1. It is unclear to me how the recursive feature elimination step would automatically remove length as a feature as sequence length is a very prominent feature in differentiating DNA vs aptamer classes (figure 2). Similarly, It is confusing that the length is still listed as a feature in table 1. I would recommend removing length feature from consideration before the feature elimination step and re-estimate the performance.

2. Similarly, was Figure 2 generated with length as input?

3. What is the number on the axis for figure 6?

4. The font is too small for figure 5,6 and 8.

5. In figure 3, the axis labels should be tsne1 and tsne2.

6. Resolution should be improved for all figures.

Reviewer #4: The manuscript is found to be properly revised as per the comments given by the reviewers. All the concerns were properly cleared and the necessary information were found to be added where ever it is required. I’m totally satisfied with the revisions made by the authors. No other or new problems were found in the manuscript.

Reviewer #5: The work is interesting in the sense of providing a new tool for discrimination of DNA aptameters using ML methods. It is well written and easy to follow. The authors have improved their work following the comments of the reviewers. I reccomned its publication in this recent form.

**Have all data underlying the figures and results presented in the manuscript been provided?**

Reviewer #1: Yes

Reviewer #2: Yes

Reviewer #4: Yes

Reviewer #5: Yes

PLOS authors have the option to publish the peer review history of their article (what does this mean?). If published, this will include your full peer review and any attached files.

Reviewer #1: No

Reviewer #2: No

Reviewer #4: **Yes: **Dr. Abilash

Reviewer #5: No

Figure Files:

Data Requirements:

Reproducibility:

References:

---

## [Editor Report · Decision Letter 2]

5 Jul 2021

Dear Mrs. Pares-Matos,

We are pleased to inform you that your manuscript 'A novel artificial intelligence-based approach for identification of deoxynucleotide aptamers' has been provisionally accepted for publication in PLOS Computational Biology.

Best regards,

Eleonora Alfinito, Ph.D.

Guest Editor

PLOS Computational Biology

Florian Markowetz

Deputy Editor

PLOS Computational Biology

---

## [Editor Report · Acceptance letter]

27 Jul 2021

PCOMPBIOL-D-20-01755R2 

A novel artificial intelligence-based approach for identification of deoxynucleotide aptamers

Dear Dr Parés-Matos,

I am pleased to inform you that your manuscript has been formally accepted for publication in PLOS Computational Biology. Your manuscript is now with our production department and you will be notified of the publication date in due course.

With kind regards,

Andrea Szabo
